# Evaluating equity dimensions of infant and child vitamin A supplementation programmes using Demographic and Health Surveys from 49 countries

Kevin Tang ![ORCID],[1,2] Hallie Eilerts,[1] Annette Imohe,[2] Katherine P Adams ![ORCID],[3] Fanny Sandalinas,[1] Grainne Moloney,[2] Edward Joy,[1] Andreas Hasman ![ORCID] [2]

[1]Department of Population Health, London School of Hygiene & Tropical Medicine, London, UK
[2]Programme Division, UNICEF, New York City, New York, USA
[3]Institute for Global Nutrition, University of California Davis, Davis, California, USA

**Correspondence to**
Dr Kevin Tang;
kevin.tang1@lshtm.ac.uk

## ABSTRACT

**Objectives** Vitamin A deficiency affects an estimated 29% of all children under 5 years of age in low/middle-income countries, contributing to child mortality and exacerbating severity of infections. Biannual vitamin A supplementation (VAS) for children aged 6–59 months can be a low-cost intervention to meet vitamin A needs. This study aimed to present a framework for evaluating the equity dimensions of national VAS programmes according to determinants known to affect child nutrition and assist programming by highlighting geographical variation in coverage.

**Methods** We used open-source data from the Demographic and Health Survey for 49 countries to identify differences in VAS coverage between subpopulations characterised by various immediate, underlying and enabling determinants of vitamin A status and geographically. This included recent consumption of vitamin A-rich foods, access to health systems and services, administrative region of the country, place of residence (rural vs urban), socioeconomic position, caregiver educational attainment and caregiver empowerment.

**Results** Children who did not recently consume vitamin A-rich foods and who had poorer access to health systems and services were less likely to receive VAS in most countries despite potentially having a greater vitamin A need. Differences in coverage were also observed when disaggregated by administrative regions (88% of countries) and urban versus rural residence (35% of countries). Differences in vitamin A coverage between subpopulations characterised by other determinants of vitamin A status varied considerably between countries.

**Conclusion** VAS programmes are unable to reach all eligible infants and children, and subpopulation differences in VAS coverage characterised by various determinants of vitamin A status suggest that VAS programmes may not be operating equitably in many countries.

## INTRODUCTION

The UNICEF Nutrition Strategy 2020–2030 commits to a goal of protecting and promoting diets, services, and practices that support optimal nutrition, growth, and development for all children to end child malnutrition in all its forms.[1] Central to achieving

---

### STRENGTHS AND LIMITATIONS OF THIS STUDY

⇒ We used Demographic and Health Survey (DHS) data from 49 low/middle-income countries to conduct a standardised analysis of the equity dimensions of vitamin A supplementation (VAS) coverage, where consistencies in findings could be broadly applied to inform global VAS policy.

⇒ The covariates included in this study, guided by the 2020 UNICEF conceptual framework, covered a diverse range of social and geographical determinants with relevance to equitable VAS programming.

⇒ The question specific to VAS in the DHS questionnaire is dependent on a multiple month recall by the caregiver of the child, which is subject to recall bias.

⇒ In settings where VAS is delivered via biannual campaigns, DHS VAS coverage may be underestimated if the timing of the survey is planned immediately before a campaign.

---

this goal requires recognising the interaction between five core systems (food, health, water and sanitation, education, social protection) to improve nutrition outcomes. However, a strategy focused on working within established systems may exacerbate unequal nutrition outcomes rooted in deeper inequities that arise from already inequitable systems.[2] Therefore, when implementing policies to ensure the most vulnerable have access to services, governments should conduct regular reviews of coverage of programmes to allow for reorientation as needed.

Vitamin A deficiency affects an estimated 29% of all children under 5 years of age in low/middle-income countries,[3] contributing to child mortality and disease burden through direct clinical manifestation (eg, xerophthalmia) and susceptibility to and exacerbated adverse outcomes from infection (eg, measles, diarrhoea).[4] For children with vitamin A deficiency or who have an increased risk of mortality, high-dose

vitamin A supplementation (VAS) can be a low-cost intervention to meet vitamin A needs when delivered using infrastructure from existing community-based delivery programmes.[5] Following an exhaustive review of the current evidence exploring the effect of VAS in preventing morbidity and mortality in infants and children,[4 6] in 2011, the WHO confirmed a recommendation for infants and children 6–59 months of age to receive a dose of VAS every 4–6 months in contexts where vitamin A deficiency is a public health problem.[7] Although the past two decades have seen increases in global VAS coverage,[8] stagnating VAS coverage in recent years brings into question whether current universal delivery strategies are consistently missing hard-to-reach children most in need, especially in settings with low coverage.[9] This is particularly the case as VAS programmes move away from delivery in campaigns towards routine delivery.[8]

The delivery of VAS can be integrated into routine community-based health service delivery programmes. In many countries, community-based delivery programmes, such as the national Essential Programme on Immunization (EPI), offer the most consistent contacts between the youngest children and the health system, thus creating a platform for the large-scale administration of VAS. However, in contrast to VAS administration protocols, childhood immunisation programmes typically benefit only children up to 1 year of age when the last vaccine dose is administered, and few countries have a contact point for immunisation at 6 months. Moreover, multi-country analyses of diphtheria–tetanus–pertussis (DTP) and measles-containing vaccines (MCV) identified gaps in programme coverage between geographical and socio-economic subpopulations.[10 11] All together, these differences suggest that the services these community-based delivery programmes provide are not equitably accessible to children with the greatest needs and may affect the VAS programmes that depend on these services for delivery.

Global guidance recommends using 'two-dose coverage' as a metric for advocacy to promote national and global progress towards achieving universal VAS coverage.[12] In this context, 'coverage' is estimated as the nationally aggregated number of VAS doses administered in a country over a 6-month period (also referred to as semester) from administrative records divided by the estimated number of children aged 6–59 months in that country for that specific semester. 'Two-dose coverage' is thus established on an annual basis as the semester in a given year with lower coverage. Two-dose coverage increases the feasibility of collating national VAS programme data biannually, which is advantageous for advocacy and other such uses. From the programmatic perspective, both two-dose coverage and single-semester coverage are limited in identifying differences in coverage among subpopulations within a country. To evaluate whether national VAS programmes are missing the children with the greatest needs, global guidance recommends using indicators generated from subnational data

that are readily available, easy to understand and relevant to the information needs of programme managers.[13]

There is a significant information gap in evaluating the equity dimensions of nutrition interventions, or the extent to which a community-based nutrition programme is reaching the children most in need.[2] Nationally representative household surveys, such as the Demographic and Health Surveys (DHS), can potentially provide insight into the equity dimensions of programme delivery. Since 1985, the DHS programme has conducted nationally representative cross-sectional surveys in over 90 predominantly low/middle-income countries, producing detailed, cross-sectional data on a variety of indicators describing demographics, health, economics and social welfare.[14] DHS questionnaires and variable nomenclature are consistent between all countries in the DHS programme, facilitating rapid analysis across multiple countries and survey years. DHS microdata enable the exploration of a variety of indicators to measure population health outcomes and the mechanisms influencing them in great depth.

DHS collect detailed information on the reception of VAS by infants and children; however, using such data to inform national VAS programmes is not straightforward. The aim of this study was to evaluate the equity dimensions of national VAS programmes according to determinants known to affect child nutrition and geographically using DHS data from multiple countries.

## METHODS
### Demographic and Health Surveys
This was a secondary data analysis using data collected as part of the DHS from various countries.[15] We reviewed DHS data from 64 countries that have been prioritised by UNICEF for support in their current national VAS programming efforts and ultimately included 49 countries in which a complete DHS was conducted since 2010, using the most recent data in countries with multiple surveys. The analysis depended on data from the Woman's Questionnaire, which contains information on the survival status of the children born to the respondent, and more detailed information on children born in the last 5 years, such as vaccination history, breastfeeding practice, recent illness and anthropometry.

The DHS protocol employs a two-stage sampling procedure.[16] In the first stage, enumeration areas are defined geographically (stratified by urban/rural residence and administrative region) using the country's most recent population census to establish the survey's primary sampling unit. In the second stage, systematic sampling identifies 20–30 households for inclusion, where selected households are visited by trained interviewers who administer the Woman's Questionnaire to women of reproductive age (15–49 years).

Our analysis included all children aged 6–59 months with available VAS history data. VAS programme participation is probed in the DHS questionnaire's *vaccination history section* by asking caregivers of children whether

their child (or children) aged 6–59 months has (or have) received a vitamin A dose in the last 6 months.[17] If possible, caregivers are asked to present proof of VAS reception using a vaccination history card, and if not available, are asked to recall VAS administration from memory after probing the caregivers to describe postnatal care activities in which they and each child have participated.[18] The primary outcome of this study was defined as the percentage of children receiving VAS in the prior 6 months, or more simply, 'VAS coverage'.

### Equity and geographical covariates

We stratified the primary outcome by several covariates describing various dimensions of equity, which were selected in accordance with the 2020 UNICEF Conceptual Framework on the Determinants of Maternal and Child Nutrition.[1] The framework used in this analysis was developed to guide nutrition programming by outlining three levels of determinants (ie, immediate, underlying, enabling) that contribute to preventing malnutrition in all its forms. By using this framework to guide covariate selection, this study recognises the levels and interconnectedness of various determinants affecting vitamin A status to define the underlying systems and processes that affect children with the greatest vitamin A needs. Geographical covariates were also selected as targeting geographically defined populations is more operationally feasible.

Covariates used to represent immediate and underlying child nutrition determinants included recent consumption of vitamin A-rich foods and access to healthcare systems and services. Recent consumption of vitamin A-rich foods from the DHS questionnaire is only collected for children aged 6–23 months, so this indicator refers predominantly to children who are within the age range where breast feeding is recommended.[19] Access to healthcare systems and services is measured using the proxy variable of vaccine reception. The first dose of the DTP vaccine (administered at 6 weeks of age) is one of the first vaccines given to an infant, where children who do not receive it are expected to have the lowest access to healthcare systems and services. Reception of the first dose of MCV (administered at 9 months of age) is expected to be delivered via similar platforms to VAS through both routine provisions and campaigns. Covariates serving as indicators for enabling child nutrition determinants included socioeconomic position, the caregiver's educational attainment and two women's empowerment dimensions calculated according to the Survey-based Women's Empowerment Index,[20] which measure the caregiver's social independence and decision-making autonomy. Geographical covariates included urban versus rural residence and administrative regions specific to each country. Descriptions of all covariates are available in table 1.

### Statistical analysis

Percentage of children receiving VAS in the prior 6 months was stratified by each equity covariate independently for every included country. All estimates were adjusted using population sample weights provided by the DHS programme.[21] Percentages of children receiving VAS in each stratum in the prior 6 months were presented with 95% CIs, where statistically significant differences in VAS coverage were identified as those with non-overlapping 95% CIs. VAS coverage by administrative region was mapped using shapefiles downloaded from GADM (V.3.6).[22]

Data analysis was conducted in RStudio (V.3.6.1, the R Foundation for Statistical Computing). Data were accessed through the DHS programme's application programming interface on 23 November 2022 via the functions available on the *rdhs* package.[23] Survey weight adjustments and statistical analyses were conducted using the functions available on the *survey* and *srvyr* packages,[24] and additional data cleaning and management used a variety of functions from the *tidyverse* package.[25]

### Patient and public involvement

As a secondary analysis of publicly available de-identified data collected as part of the DHS, patients and the public who were included as part of the study population were not involved in the design of this study.

## RESULTS

### Summary of included surveys and populations

In total, the birth history records of 1 465 369 women, corresponding to 608 388 children 6–59 months, were included in this study. For the survey question asking about VAS reception within the prior 6 months, the rate of response was >90% for all countries included. A summary of all included countries and the DHS data used in this study is available in online supplemental table 1.

The national VAS coverage mean among included countries was 58% and ranged from 28% to 83%, where countries with the lowest national VAS coverage were Papua New Guinea (28%), Haiti (28%), Kyrgyzstan (39%) and Guinea (40%). The percentage of children who did not recently consume vitamin A-rich foods among included countries ranged from 19% to 74%, where countries with the lowest percentage of consumption were Burkina Faso (74%), Niger (73%) and Ethiopia (70%). For vaccinations, the percentage of non-vaccinated children among included countries ranged from 1% to 48% for DTP1 and 13% to 69% for MCV1. A descriptive summary of the survey population of children aged 6–59 months for each country is provided in online supplemental table 2.

In figures 1 and 2, we present VAS coverage for all countries stratified by recent consumption of vitamin A-rich foods and access to healthcare systems and services. VAS coverage was significantly higher among children who recently consumed vitamin A-rich foods, compared with children who did not in 85% (n=41) of countries (figure 1). For access to healthcare systems and services, VAS coverage was significantly higher in children who had access to their first dose of the DTP vaccine in 47 of

**Table 1** Description of study covariates used to represent immediate, underlying and enabling determinants of child nutrition and geography

| Determinant | Type | Coding details |
|---|---|---|
| Recent consumption of vitamin A-rich foods | Immediate/underlying | Whether the following types of food were reported being consumed by the child (aged 6–23 months) in the 24 hours prior to the interview: eggs, any meat, fish, orange-fleshed pumpkin/carrot/squash/sweet potato, vitamin A-rich fruit (mango, papaya), liver/heart/organ meat, dark green leafy vegetables |
| Access to healthcare services and systems (DTP1 vaccine) | Immediate/underlying | Whether the child received the first dose of the DTP vaccine, administered at 6 weeks of age per international guidelines |
| Access to healthcare services and systems (MCV1 vaccine) | Immediate/underlying | Whether the child received the first dose of the MCV, administered at 9 months of age per international guidelines |
| Administrative region | Geography | Administrative region the household is located, defined by the subnational reporting area (provinces or groups of provinces) as defined by the DHS recode |
| Place of residence | Geography | Whether household is in an urban or rural location |
| Socioeconomic position | Enabling | Quintile of household wealth defined using a principal component analysis score comprised of household living conditions and durable assets (poorest=1st quintile, wealthiest=5th quintile) |
| Caregiver's social independence | Enabling | Women's empowerment indicator calculated as quintiles of SWPER Index. Social independence domain included data related to education, frequency of reading newspapers/magazines, and age at first childbirth and at first cohabitation (includes children with partnered caregivers) (least socially independent=1st quintile, most socially independent=5th quintile) |
| Caregiver's decision-making autonomy | Enabling | Women's empowerment indicator calculated as quintiles of SWPER Index. Decision-making autonomy domain included questions about involvement in household decisions and whether the respondent worked in the past 12 months (includes children with partnered caregivers) (least decision-making autonomy=1st quintile, most decision-making autonomy=5th quintile) |
| Caregiver's educational attainment | Enabling | Highest educational attainment of caregiver (none, primary school, secondary school or higher) |

DHS, Demographic and Health Survey; DTP, diphtheria, tetanus and pertussis; MCV, measles-containing vaccine; SWPER, Survey-based Women's Empowerment.

the 49 countries compared with those who did not have access to these vaccines (figure 2A). VAS coverage was significantly higher in children who had access to their first dose of MCV in all 49 countries (figure 2B).

### VAS coverage by geography and other enabling determinants

In figure 3, VAS coverage is stratified by geographical covariates of vitamin A nutritional status for all countries. For place of residence, 31% of countries (n=15) had significantly lower VAS coverage in populations residing in rural versus urban areas. In countries where VAS coverage in rural residences exceeded that of urban residences, differences in coverage were never >8%. For administrative region, regions with the lowest and highest VAS coverage in each country had significant differences in 88% of countries (n=43) with spatial structure in subnational VAS coverage visible when mapped (figure 4 for Chad, India, Nigeria, Ethiopia, Yemen; full set of country maps available in online supplemental figures 1–5).

In 73% of countries (n=36), VAS coverage was significantly higher in the wealthiest quintile of the population compared with the poorest quintile of the population, where the differences in coverage between poorest and wealthiest were greatest in Nigeria (33% difference),

Cote D'Ivoire (25% difference) and the Democratic Republic of the Congo (25% difference) (online supplemental figure 6). In 37% of countries (n=18), children of caregivers who were the most socially independent had significantly higher VAS coverage compared with caregivers who were the least socially independent (online supplemental figure 7). In 20% of countries (n=10), VAS coverage in children of caregivers who were most autonomous in their decision-making was significantly higher compared with children of the least autonomous caregivers (online supplemental figure 8). Caregivers with higher educational attainment had higher VAS coverage for their children compared with caregivers with lower educational attainment in several countries (online supplemental figure 9).

### DISCUSSION

This study used open-source data to identify inequities in VAS programme coverage so that strategies can be devised to improve VAS coverage among unreached populations. We found that children who likely have lower access to vitamin A-rich foods and who have impaired access to

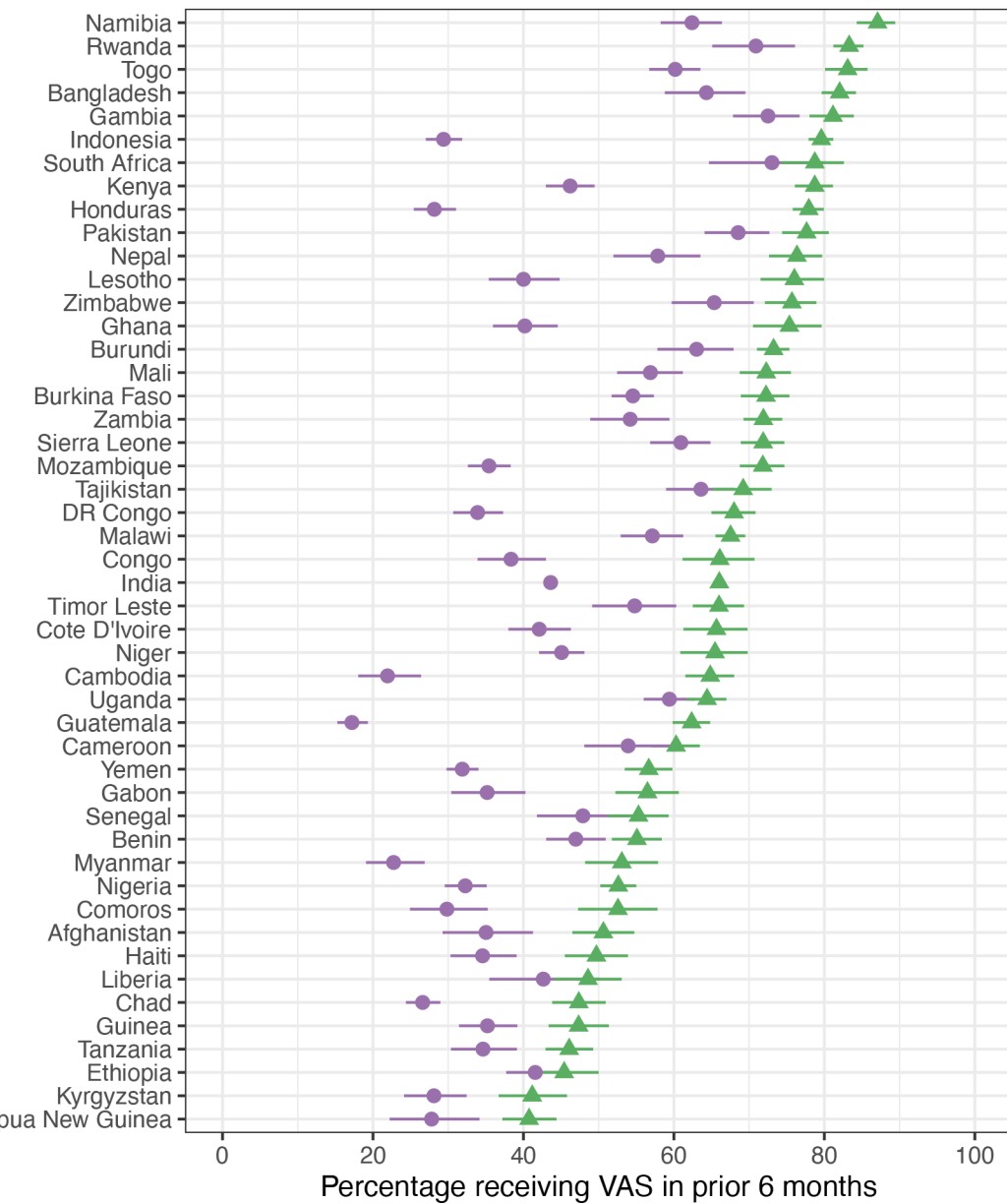

**Figure 1** Vitamin A supplementation (VAS) coverage among children 6–23 months who have and have not recently consumed vitamin A (VA)-rich foods (in the 24 hours prior to the interview). Philippine Demographic and Health Survey did not collect data on child consumption of VA-rich foods.

healthcare systems and services are also less likely to receive VAS. This pattern was consistent across most countries and highlights the challenges current VAS delivery faces to reach children who are likely most in need of VAS. The analysis also shows that in some countries, children missed by VAS also reside in the poorest households, in rural areas and have caregivers who are more constrained by gender norms. Although these are the general trends, we also observed considerable variation between countries. While the use of DHS data to gain insight into VAS programmes has been conducted in some countries,[26–28] future analyses using DHS data can help inform VAS programme operations in other country

contexts. National programmes can use the analysis presented here as an advocacy tool for universal coverage, or as a framework to improve targeting and prioritisation of children who are likely to be most in need of VAS delivery programmes.

The demonstrated use of DHS data to provide subnational equity perspectives can provide useful VAS programme insights. For example, this study indicated that lower VAS coverage for children who may have lower access to vitamin A-rich foods and healthcare systems and services was consistent across most countries. According to global estimates using UNICEF infant and young child feeding data, the consumption of vitamin A-rich foods by

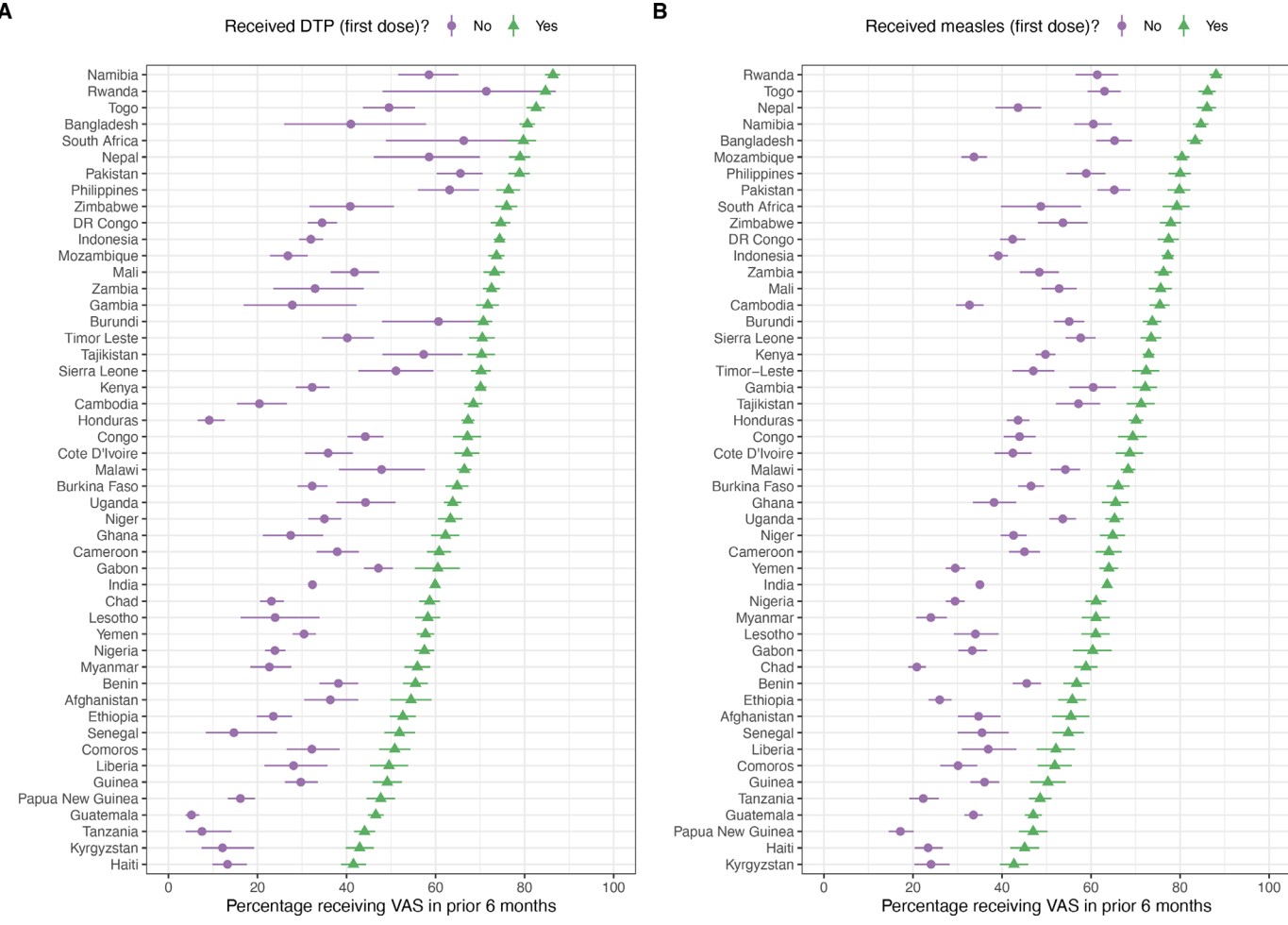

**Figure 2** Vitamin A supplementation (VAS) coverage among children who have and have not received (A) the first dose of the diphtheria–tetanus–pertussis (DTP) vaccine and (B) the first dose of the measles-containing vaccine (indicator of access to healthcare systems and services).

complementary feeding children is lowest in poorer and more rural children[29] suggesting that in combination, multiple enabling determinants affecting nutrition are likely to contribute to reduced access to vitamin A-rich foods. However, this study suggests that in many countries, VAS coverage is either equal or lower in these rural and poorest populations despite likely having greater need for VAS.

To address these gaps, VAS programmes may benefit from being more aligned with routine community-based delivery programmes. In children aged 6–11 months, in countries where VAS coverage between vaccinated and unvaccinated children diverges, there is a potential opportunity to improve VAS coverage by aligning programmes more closely to the country's EPI systems. This alignment with EPI systems will require programme reforms, including the creation of a 6-month contact point in routine systems, inclusion of VAS dosing schedules on child health cards,[30] and strengthening supply chains for vaccines and vitamin A capsules to ensure concomitant availability. For older children who are not serviced by the EPI (12–59 months), there may be other established routine early childhood programmes where the routine

delivery of VAS could be integrated, such as growth monitoring and promotion, counselling on breast feeding and complementary feeding, early childhood development programmes and the early detection and treatment of severe wasting.[31] Current reliance on polio vaccination campaigns to deliver VAS has presented a risk of declined VAS coverage as polio programmes cease,[32] so identifying other routine community-based delivery programmes to integrated VAS should be prioritised.

As the global vitamin A landscape evolves, it is important for national governments to consider how VAS programmes can be positioned in combination with other parallel vitamin A interventions to reduce risks of deficiency for all children. VAS has short-term benefits for children (boosting serum retinol for approximately 2 months after administration),[33] so other interventions are necessary to sustainably maintain adequate vitamin A intake through the diet. For the countries included in this study, 51% (n=25) have nationally mandated the large-scale vitamin A fortification of industrially produced food items.[34] However, poorer and rural populations—where VAS coverage is often lower—often consume small quantities of fortified food items (eg, cooking oil, sugar, wheat

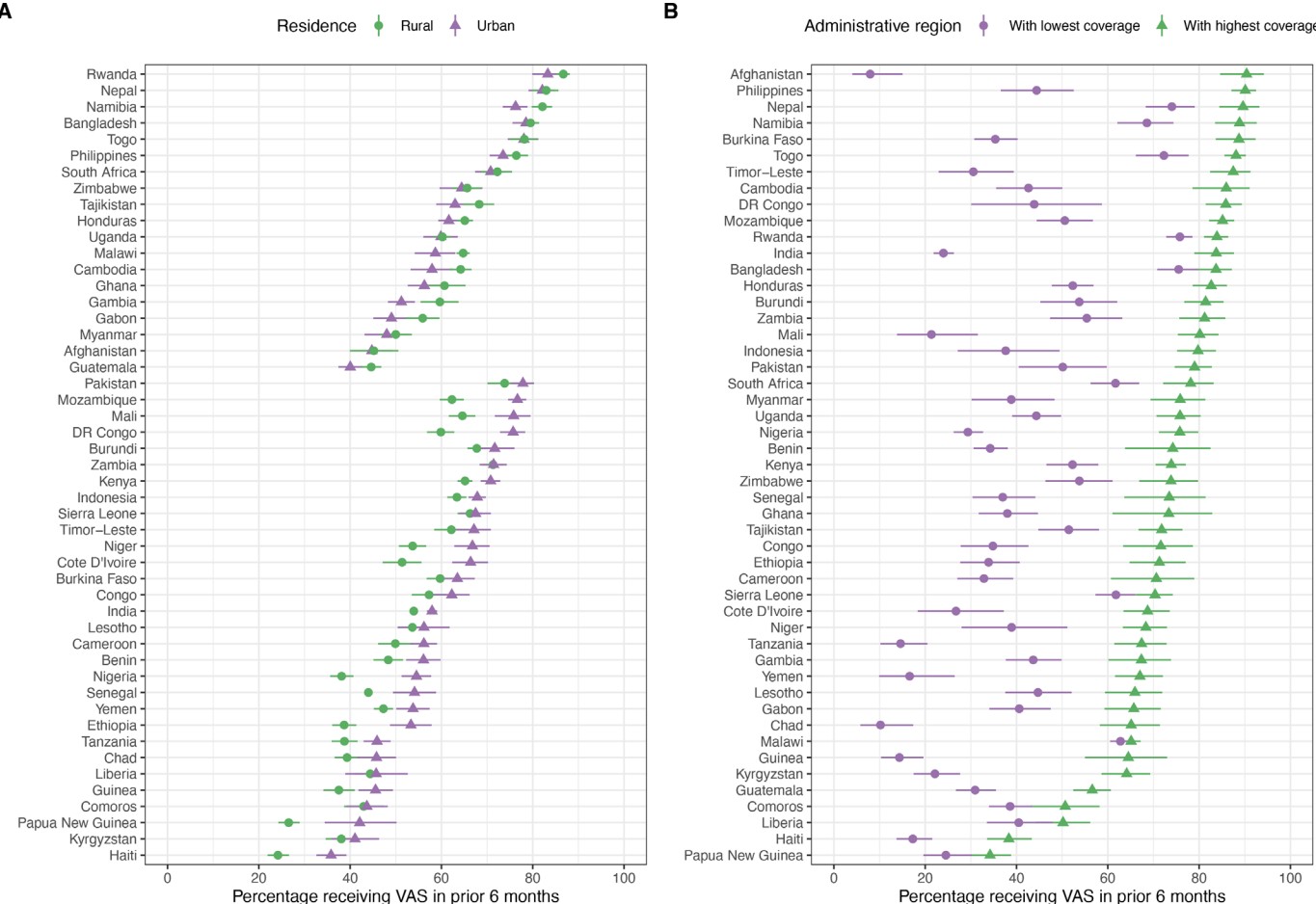

**Figure 3** Vitamin A supplementation (VAS) coverage disaggregated by geography: (A) rural versus urban residence and (B) administrative regions.

flour) and may therefore derive limited benefit from industrial vitamin A fortification schemes.[35] Programmes aimed at broader improvements in dietary diversity have also been recommended,[36] but variation in diets between and within countries poses challenges when scaling up across different contexts. With multiple parallel vitamin A interventions, future studies that simultaneously consider the individual and combined contributions of all vitamin A interventions could help governments orient their national vitamin A strategies.

Evaluating equity dimensions of VAS programmes by stratifying coverage estimates using multiple variables can help identify potential gaps in national programmes' delivery strategies. This analysis drew from the DHS Household and Woman's Questionnaires to identify enabling determinants to undernutrition that could affect VAS coverage. If differences in VAS coverage are detected between populations, some of these geographical determinants (eg, place of residence, administrative regions of the country) can serve as indicators that lead to practical recommendations for programmes to adopt to bridge coverage gaps (eg, strengthening programmes in a specific region of the country or for a specific demographic). In contrast, other determinants, while useful to evaluate whether VAS programmes are broadly equitable,

lack clear operational directions on how programmes can specifically target populations that are left behind (eg, socioeconomic position, caregiver's social independence, caregiver's decision-making autonomy). This study explored several different indicators made available in the DHS, and VAS programmes could benefit from further research aimed at interpreting the combination of multiple indicators in the context of a country's current VAS delivery strategy (eg, campaign vs routine delivery).

In our analysis, there are several limitations that are important to consider. First, the question specific to VAS in the DHS questionnaire is dependent on recall by the caregiver of the child. The percentage of respondents who could prove VAS reception using home-based vaccination records varied between countries, but for most countries (69%; n=34), less than half of respondents had proof of VAS reception (online supplemental figure 10). For children whose records depend solely on recall by the caregiver, recall error (eg, confusing VAS droplets with polio droplets, imprecision in the exact date of reception, confusion between multiple children) is more likely. Second, in settings where VAS is delivered via biannual campaigns, interpretation of DHS VAS coverage must cautiously consider the timing of the survey relative to the timing of the campaign (eg, VAS coverage

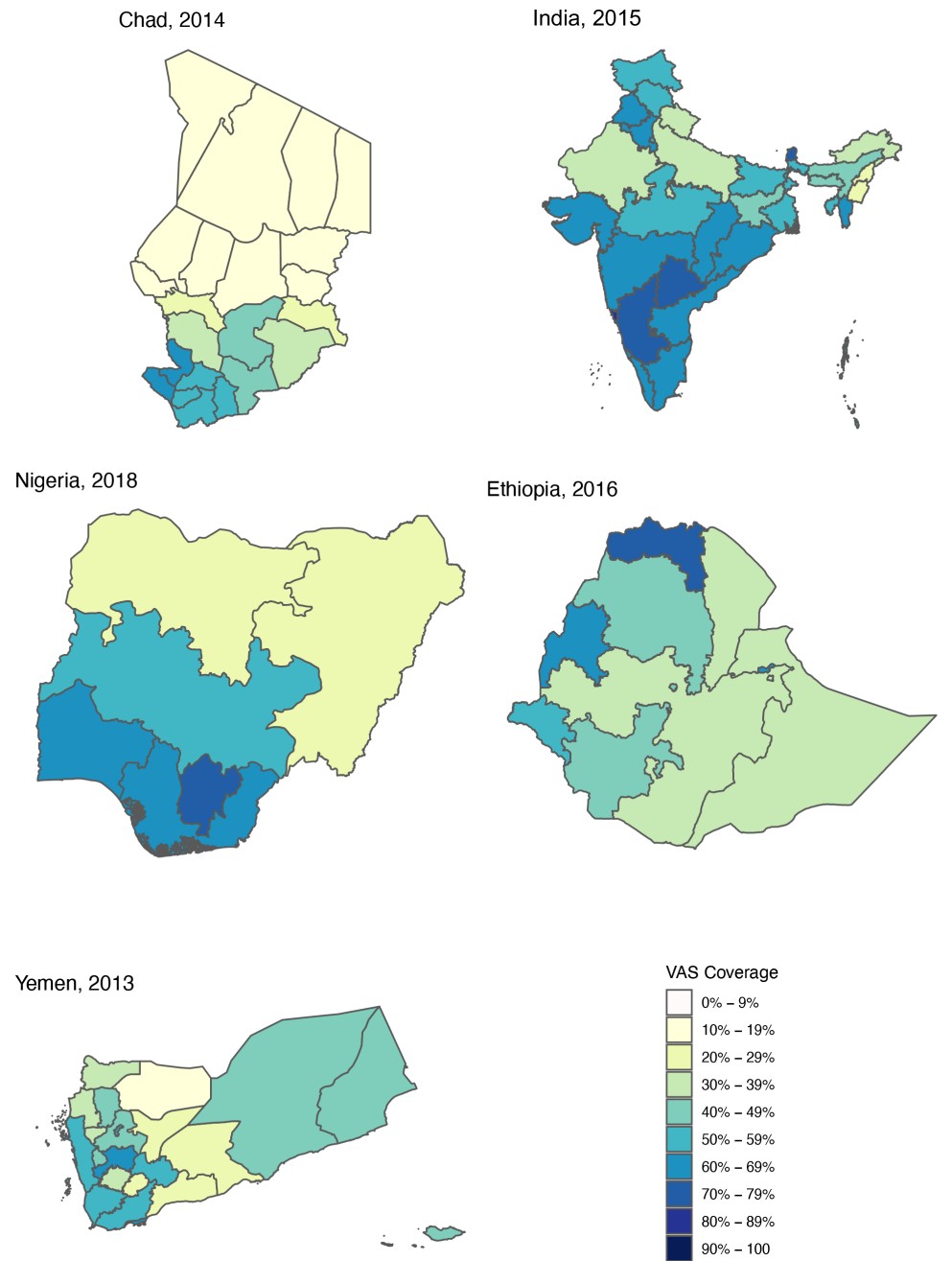

**Figure 4** Vitamin A supplementation (VAS) coverage maps by administrative regions (maps of all countries available in the online supplemental material 1).

may be underestimated if the DHS was implemented immediately before the start of a new campaign). Across several settings, the characteristics of children missed by current VAS programmes remained relatively consistent, but recommendations related to specific settings would benefit from country-specific interpretation to place this information in the context of a country's current VAS policies and programmes. Third, the dietary data collected as part of the DHS only contain one dichotomous recall of whether vitamin A-rich foods were consumed in the past 24 hours by complementary feeding children. No information is available regarding food consumption quantities, weekly variation in diets or for children aged 24–59 months. With this kind of dietary data, it is not possible to understand whether VAS is being administered to children who have inadequate dietary vitamin A intake or who have intake that exceeds daily upper limits to put children at risk of toxicity. To fully understand whether VAS programmes are addressing children with the greatest vitamin A needs, VAS coverage should be estimated alongside other subnational nutritional assessment data (eg, vitamin A inadequacy from dietary assessment) to identify populations with both inadequate dietary vitamin A intake and low VAS coverage. Considering these limitations, VAS coverage data available as part of the DHS are not recommended to replace

either single-semester coverage, two-dose coverage or efforts to advocate for additional repeated, representative surveys. However, DHS data analysed in parallel to these other available data can provide a more complete understanding of the VAS context in a country to inform governments on national programme performance and help characterise missed children so that strategies can be adapted to reach them.

## CONCLUSION

This analysis further contributes clear and consistent evidence that VAS programmes are unable to reach all eligible infants and children. This analysis also highlighted inequity in access to VAS, as children who are likely the most in need are more often not reached by this life-saving intervention. Three decades after the WHO first recommended high-dose vitamin A supplements to infants and children aged 6–59 months, countries with high risks of vitamin A deficiency have not yet achieved the goal of universal VAS coverage. Particular attention should focus on settings with low coverage, where children who are the most in need are also more likely to be missed. While DHS data can be useful to identify variations in VAS coverage based on equity dimensions already included in the questionnaire, challenges remain for programmatic questions requiring current, context-specific data.

**Contributors** KT, KPA and AH designed the study. KT, AI, KPA and AH defined the variables for disaggregation. KT and HE wrote the code for data extraction and analysis. FS, GM, EJ and AH contributed to the contextualisation of analysis results to the broader field of research and policy. All authors critically reviewed and approved the final manuscript. KT acts as the guarantor of this work and accepts full responsibility for the work.

**Funding** No funding was provided for this study. KT, KPA, FS and EJ receive funding from the Bill and Melinda Gates Foundation through the Micronutrient Action Policy Support (MAPS) project (INV-002855). KT, AI, GM and AH are staff members at UNICEF.

**Disclaimer** The contents of this article do not necessarily reflect the views of the Bill and Melinda Gates Foundation or UNICEF.

**Map disclaimer** The inclusion of any map (including the depiction of any boundaries therein), or of any geographic or locational reference, does not imply the expression of any opinion whatsoever on the part of BMJ concerning the legal status of any country, territory, jurisdiction or area or of its authorities. Any such expression remains solely that of the relevant source and is not endorsed by BMJ. Maps are provided without any warranty of any kind, either express or implied.

**Competing interests** None declared.

**Patient and public involvement** Patients and/or the public were not involved in the design, or conduct, or reporting, or dissemination plans of this research.

**Patient consent for publication** Not required.

**Ethics approval** This work is a secondary analysis of publicly available, de-identified data collected as part of the DHS. No Ethics Committee or Institutional Board approval is required.

**Provenance and peer review** Not commissioned; externally peer reviewed.

**Data availability statement** Data are available in a public, open access repository. All data used in this study are made available by the DHS programme on https://www.dhsprogram.com/.

**ORCID iDs**
Kevin Tang http://orcid.org/0000-0003-2580-3726
Katherine P Adams http://orcid.org/0000-0002-1060-2473
Andreas Hasman http://orcid.org/0000-0002-0431-2947

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
