## [Reviewer comments · BMJ Open]

ARTICLE DETAILS

TITLE (PROVISIONAL)	Evaluating equity dimensions of infant and child vitamin A supplementation programs using Demographic and Health Surveys from 49 countries
AUTHORS	Tang, Kevin; Eilerts, Hallie; Imohe, Annette; Adams, Katherine; Sandalinas, Fanny; Moloney, Grainne; Joy, Edward; Hasman, Andreas

VERSION 1 – REVIEW

REVIEWER	Greig, Alison Nutrition International, Global Technical Services
REVIEW RETURNED	04-Apr-2022

GENERAL COMMENTS	This paper is well done, and the subject is timely and relevant. I only have a few comments, mainly with respect to the recommendations and discussion of the results. Comments are inserted in the attached file - The reviewer provided a marked copy with additional comments. Please contact the publisher for full details.
--

REVIEWER	Kruger, Iolanthe
REVIEW RETURNED	05-Apr-2022

GENERAL COMMENTS	Some of the references used are outdated - older than five years. The authors are requested to revisit the references and use more recent references.
---

REVIEWER	Vidaletti, Luis Federal University of Pelotas, International Center for Equity in Health
REVIEW RETURNED	03-Oct-2022

GENERAL COMMENTS	Thank you for the opportunity to review and be able to contribute to science with this manuscript. ABSTRACT Objectives Line25: “low- and lower middle-income countries” is not correct. “low- and middle-income countries” is the correct, since Gabon (2012), Namibia (2013), and South Africa (2016) are middle-income countries. INTRODUCTION Line 80: “xerophthalmia” is a typo. The correct is “xerophthalmia”. Line 124:
---

	“low-middle income countries”. The usual term is “low-and middle-income countries”. METHODS Statistical Analysis Line 201: “VAS coverage by administrative region was mapped using shapefiles downloaded from GADM (version 3.6)” missing reference, please provide us with a reference for this. RESULTS Summary of included Surveys & populations Lines 221-223: By the https://www.statcompiler.com/en/, the national VAS coverage mean for the same sample in this study was 61% and ranged from 30% to 86%, where countries with the lowest national VAS coverage were Haiti (30%), Papua New Guinea (31%), Guinea (41%), Tanzania (41%), and Kyrgyzstan (44%). Table of Contents Supplementary Table 1. Countries included in the study and summary information on their respective Demographic and Health Surveys Line 28: “India 2012” I think it’s a mistake. India does not have a DHS survey in 2012 but in 2015-16. Why were surveys such as Angola 2015, Dominican Republic 2013, Egypt 2014, Jordan 2017, and Peru 2012 not included in the study? And why for countries like Ethiopia, Rwanda, and Senegal did you use the 2016, 2015, and 2018 surveys (respectively) and not the most recent 2019 surveys?
--	--

VERSION 1 – AUTHOR RESPONSE

Reviewer 1: Ms. Alison Greig, Nutrition International

Comment 1:

This paper is well done, and the subject is timely and relevant. I only have a few comments, mainly with respect to the recommendations and discussion of the results. Comments are inserted in the attached file

Response: We thank the reviewer for the positive feedback and we appreciate their contributions to helping improve our work.

Reviewer 1: Ms. Alison Greig, Nutrition International (INTRODUCTION)

Comment 2:

Line 75: *Do the authors mean regular review by governments? The section above is referring to the UNICEF strategy - so it is unclear if it is meant for UNICEF to regularly review, or if this means governments.*

Response: We thank the reviewer for identifying this point for further clarification. Yes, in this statement, we argue that when implementing national policies, it is important that government programs review their progress to ensure that the most deprived populations with the greatest

needs are benefitting from these government programmes. To clarify this point, this statement was rephrased as follows:

“Therefore, when implementing policies to ensure the most vulnerable have access to services, governments should conduct regular reviews of coverage of programmes to allow for reorientation as needed.”

Comment 3:

Are the authors recommending that a targeted approach is needed due to the global drops in coverage? The Fawzi Wang paper that is referenced questions if universal delivery is still needed due to less burden of VAD/measles/U5MR - they are not suggesting targeting to make up for dropped coverage. Perhaps this point can be reviewed? I think the nuance is that universal approach can miss those most in-need, especially in settings with low coverage.

Response: We thank the reviewer for highlighting this important point about programme targeting. Fawzi and Wang do suggest potentially initiating a targeted approach for VAS programmes (paragraphs 3 and 11)¹, but this is recommended in the context of having available repeated, representative surveys. While we encourage national governments to use available data to contextualise gaps in coverage, we recognise that there are limitations to DHS data on VAS coverage that limit its use for programme targeting, such as long periods between survey implementation years. We have modified the text throughout to de-emphasise the use of DHS data for programme targeting and highlighting the nuance proposed in the reviewer’s recommendation.

Comment 4:

The WHO website reference (10) is recommending that VAS should be integrated into immunization contact points via routine health system and/or campaigns. It does not discuss community-based. Perhaps it is a matter of just removing the reference?

Response: We thank the reviewer for noticing this. We removed the reference and changed the sentence to say that VAS can be integrated into community-based health programs. The statement is now phrased as follows:

“The delivery of VAS can be integrated into routine community-based health service delivery programs. In many countries, community-based delivery programs, such as the national Essential Programme on Immunization (EPI), offer the most consistent contacts between the youngest children and the health system...”

Comment 5:

The globally accepted indicator for tracking global progress in VAS programs is "two-dose coverage" however two dose coverage is a computation, and is not the recommended indicator for Monitoring VAS programs. The global guidance for monitoring VAS programs is as per the GAVA Guides on Monitoring VAS authored by NI, UNICEF, HKI and CDC -- which makes the strong recommendation to monitor semester by semester and only compute 2-dose coverage if needed for advocacy or other such uses.

Response: We thank the reviewer for highlighting this important reference and we agree that it is important to emphasise that two-dose coverage is intended for the purposes of advocacy and higher-level messaging. To reflect this, we have modified the text to clarify the use of two-dose coverage for the purpose of advocacy. Also, we have referenced the GAVA document, Monitoring of Vitamin A Supplementation: A Guide for National Programme Managers, in the manuscript text referring to two-dose coverage. The manuscript text now reads as follows:

“Global guidance recommends using “two-dose coverage” as a metric for advocacy to promote national and global progress towards achieving universal VAS coverage. In this context, “coverage” is estimated as the nationally aggregated number of VAS doses administered in a country over a six-month period (also referred to as semester) from administrative records divided by the estimated number of children ages 6 – 59 months in that country for that specific semester. “Two-dose coverage” is thus established on an annual basis as the semester in a given year with lower coverage. Two-dose coverage increases the feasibility of collating national VAS program data bi-annually, which is advantageous for advocacy and other such uses.”

Comment 6:

This point here is exactly why 2-dose coverage is not recommended by GAVA for use at the national level. GAVA guidance recommends looking at subnational data, as well as looking at coverage by semester only.

Response: We thank the reviewer for highlighting indicators specifically relevant for VAS programming compared to those that are relevant for national advocacy. To best reflect the differences between indicators used for national and global advocacy versus indicators used for subnational programming, we have referenced the GAVA document entitled “Monitoring of Vitamin A Supplementation: A Guide for District (Area-based) Programme Managers” and included in text more nuance referring to subnational-level program monitoring. The manuscript text now reads as follows:

“From the programmatic perspective, both two-dose coverage and single semester coverage are limited in identifying differences in coverage among sub-populations within a country. To evaluate whether national VAS programs are missing the children with the greatest needs, global guidance recommends using indicators generated from subnational data that are readily available, easy to understand, and relevant to the information needs of program managers.”

Reviewer 1: Ms. Alison Greig, Nutrition International (RESULTS)

Comment 7:

There is likely also something to say about how this varies in settings with overall low national coverage and also how this varies in the settings where only 25-30% of children 6-23 months of age consumed VA-rich foods such as BF, ET etc. Was this analysis done?

Response: We thank the reviewer for bringing up this opportunity for additional analyses in our work. The aim of our analysis was to provide an understanding for how VAS coverage estimates from DHS data could be used to understand the equity implications of national VAS programs. We designed our methodological approach to demonstrate broad patterns in coverage differences between several determinants known to affect child nutrition. While there are several future research opportunities to

further dissect some of these determinants in individual countries or groups of countries, we found this to be out of scope for this broad level analysis. We do recommend in our discussion further research interpreting these determinants in combination for more nuanced recommendations to a particular country's national strategy.

Reviewer 1: Ms. Alison Greig, Nutrition International (DISCUSSION)

Comment 8:

Noting that at the end of this section some limitations are noted about use of DHS data, however suggest the discussion also mention the limitation of DHS data to estimate coverage in settings where VAS is delivered via biannual campaigns, as is the case for approximately more than half of the countries included in this analysis. Not only is caregiver recall a limitation, it is also the timing of the data collection relative to the timing of the campaigns as the question asks "in the last 6 months" If the next campaign is about to begin in the coming month the coverage may be underestimated - this would especially affect the analysis comparing VAS coverage and access to routine services -- as settings where routine services are weak are often the ones delivering VAS via campaigns.

Response: We thank the reviewer for raising this important limitation and we agree that this point should be included in our list of limitations to consider when using DHS data to evaluate VAS programs. This limitation was included in the main manuscript text as follows:

“Second, in settings where VAS is delivered via biannual campaigns, interpretation of DHS VAS coverage must cautiously consider the timing of the survey relative to the timing of the campaign (e.g., VAS coverage may be underestimated if the DHS was implemented immediately before the start of a new campaign). Across several settings, the characteristics of children missed by current VAS programs remained relatively consistent, but recommendations related to specific settings would benefit from country-specific interpretation to place this information in the context of a country's current VAS policies and programs.”

Comment 9:

Line 372: I would argue it doesn't tell us how to reach them - it tells us who the unreached are, so that strategies might be able to be adapted to try to reach them.

Response: We thank the reviewer for recommending this adaptation to our argument. In line with the revisions addressed above to orient the manuscript text to place less emphasis on program targeting, we have also revised this statement. As recommended by the reviewer the text now emphasize that DHS VAS data is more applicable for characterising unreached children, as follows:

“However, DHS data analyzed in parallel to these other available data can provide a more complete understanding of the VAS context in a country to inform governments on national program performance and help characterize missed children so that strategies can be adapted to reach them.”

Comment 10:

Given what the authors mention above about the limitations of available capacity to conduct detailed re-analysis of DHS data, infrequent DHS surveys that may limit timely data - is this study

recommending governments do this sort of analysis repeatedly to inform their programs - or does this study provide the additional evidence to support what we know which is VAS programs may be missing those who need it the most. I think instead the recommendation to governments is not redo this analysis, as the findings are clear and consistent across many settings - instead the recommendation to governments is to use this evidence base as reason to pursue two possible program solutions: 1) do more to strive for universal coverage, because this increases the chances of reaching the full spectrum of children; or 2) in settings where coverage is low it is even more imperative to target the VAS to those most in need because the 50% being reached may be those who need it the least. The challenge and severe limitation to this solution is to identify what data is readily available to inform targeting on an individual basis.

Response: We thank the reviewer for this important contribution. We have carefully considered your comments and we agree with the broader message recommended. We therefore modified the general message and primary recommendation of this analysis. We made changes throughout the manuscript to reflect the use of this analysis for advocacy rather than recommending conducting similar analysis. The changes are as follows:

Line 43-44: The conclusion of the abstract was modified to maintain consistent messaging.

Line 275-277: We modified the high-level discussion summary to state “National programs can use the analysis presented here as an advocacy tool for universal coverage, or ...”

Lines 342-353 (original manuscript): We removed this section in our discussion that recommends the subsequent analysis of DHS data in routine monitoring. We are not suggesting redoing this type of analysis as we recognise that continuing these activities is dependent on continuous, repeated DHS surveys and is not standard for all countries barring a few exceptions.

Line 338 – We included this simple change of wording to highlight the limitations of our analysis rather than flagging them as considerations for future analysts repeating this analysis. It is now phrased as the following: “In our analysis, there are several limitations that are important to consider.”

Line 379-382 (original manuscript): We removed the final limitation in the original submission as it was less relevant in a message focusing on using the results of this data for universal VAS advocacy.

Line 370-379: We have rewritten the conclusion so that the text is consistent with these high-level advocacy points.

Reviewer 2: Iolanthe Kruger

Comment 1:

Some of the references used are outdated - older than five years. The authors are requested to revisit the references and use more recent references.

Response: We thank the reviewer for contributing this comment to improve our study. We have included some additional references that refer to the most current global guidance as also recommended by the previous reviewer. We do opt to keep many of the older references that were included in our original submission. Vitamin A supplementation has been recommended for infants and children aged 6-59 months for the past 30 years and the DHS program has been operating in select countries since 1984. While these entities have

undergone development throughout those periods, earlier published research and guidance are still very relevant to the current discussions explored in this study.

Reviewer 3: Dr. Luis Vidaletti, Federal University of Pelotas

Comment 1:

Thank you for the opportunity to review and be able to contribute to science with this manuscript.

Response: We thank the reviewer for their thorough review and useful contributions to our work.

Reviewer 3: Dr. Luis Vidaletti, Federal University of Pelotas (ABSTRACT)

Comment 2:

Line 25: *“Low- and lower middle-income countries” is not correct. “low- and middle-income countries” is the correct, since Gabon (2012), Namibia (2013), and South Africa (2016) are middle-income countries.*

Response: We thank the reviewer for this comment. We have corrected the text to use “low- and middle-income countries” consistently throughout the manuscript.

Reviewer 3: Dr. Luis Vidaletti, Federal University of Pelotas (INTRODUCTION)

Comment 3:

Line 80: *“xerophthalmia” is a typo. The correct is “xerophthalmia”.*

Response: We thank the reviewer for pointing out this error. We have corrected this spelling mistake in the revised manuscript text.

Comment 4:

Line 124: *“low-middle income countries”. The usual term is “low-and middle-income countries”.*

Response: We thank the reviewer for flagging this correction. We have corrected the terms as recommended.

Reviewer 3: Dr. Luis Vidaletti, Federal University of Pelotas (METHODS)

Comment 5:

Line 201: *“VAS coverage by administrative region was mapped using shapefiles downloaded from GADM (version 3.6)” missing reference, please provide us with a reference for this.*

Response: We thank the reviewer for this comment. We have added a reference to this statement as recommended.

Comment 6:

Lines 221-223: *By the <https://www.statcompiler.com/en/>, the national VAS coverage mean for the same sample in this study was 61% and ranged from 30% to 86%, where countries with the lowest national VAS coverage were Haiti (30%), Papua New Guinea (31%), Guinea (41%), Tanzania (41%), and Kyrgyzstan (44%).*

Response: We thank the reviewer for checking the results from our national-level results presented in Supplementary Table 2 to the same indicator calculated by StatCompiler. In Table 1 below, we have presented this study’s estimated VAS coverage alongside the corresponding StatCompiler estimate for the countries queries and the absolute difference between them. We constructed our indicators using the standardised methods recommended by the DHS Program², and believe the discrepancies are minor.

The aim of our study was to understand the equity dimensions of national VAS programs according to various determinants. While we recognise the importance of the national VAS estimate, we believe that our estimates were properly calculated to support the main aim of our analysis, which was to stratify estimates of coverage by various equity determinants. We prefer to use the estimates calculated by this study to populate Supplementary Table 1 instead of the estimates produced by StatCompiler as our estimates serve as the foundation for the rest of this study.

Table 1. Comparison of VAS coverage estimated in this study and by StatCompiler.

Country	Study estimate, % (95%CI)	StatCompiler estimate	Absolute difference
Haiti	28.3 (26.3, 30.4)	30.0	1.7
Papua New Guinea	28.2 (25.9, 30.6)	30.7	2.5
Guinea	40.0 (37.3, 42.6)	41.1	1.1
Tanzania	40.6 (38.4, 42.9)	41.3	0.7
Kyrgyzstan	39.0 (36.1, 41.9)	43.8	4.8

Comment 7:

Line 28: *“India 2012” I think it’s a mistake. India does not have a DHS survey in 2012 but in 2015-16.*

Response: We thank the reviewer for noticing this error in labelling the year of the India DHS survey. We have corrected the labelling throughout the text.

Comment 8:

Why were surveys such as Angola 2015, Dominican Republic 2013, Egypt 2014, Jordan 2017, and Peru 2012 not included in the study?

Response: We thank the reviewer for this important question on the methods for country selection used in this study. When selecting countries for inclusion, our aim was to use data from a broad list of countries to link consistencies in results across various settings, but also ensure that our recommendation were specific enough for relevance to national level VAS policies.

First, we reviewed a list of 64 countries that have been prioritised by UNICEF for support in their current national VAS programming efforts (Figure 1). UNICEF has invested considerable efforts in these 64 countries by providing governments with data and linking discussions to guidance, which has helped establish an adequate foundation for engaging with national VAS policies and programs in these settings³. Second, in these priority countries, we searched the DHS API for any surveys that had been conducted since 2010 and included all countries with available data. The variables we used in DHS recodes before 2010 have been updated and were not consistent with recent surveys that were the focus for this study. We have revised this description in our methods to improve clarity.

Angola was the only country noted in the comment that fulfilled all inclusion criteria. However, several variables of interest, including VAS reception in the past 6 months, were not accessible when retrieving the Angola 2015 DHS data from the API and therefore it could not be included.

Including other non-UNICEF priority countries with recent DHS data in this study was explored, but ultimately not pursued since non-priority countries have varying policies and positions related to VAS that make the relatively ridged structure of DHS questionnaires challenging to interpret. In Egypt for example, the Ministry of Health and Population's policy is to provide VAS at age 9 - 18 months, so coverage would require further interpretation with respect to the age distribution of the sample. In Peru, VAS coverage was extremely low (<5%), and questions about data quality or political priority of VAS could not be explored further since UNICEF does not have semester or two-dose coverage data to compare in this setting. Due to these challenges, we decided to focus only on the UNICEF priority countries.

Comment 9:

And why for countries like Ethiopia, Rwanda, and Senegal did you use the 2016, 2015, and 2018 surveys (respectively) and not the most recent 2019 surveys?

Response: We thank the reviewer for highlighting this important observation. When we submitted this study to the journal, we used the most recent data available on 1 February 2022 (near the time of our submission to the journal), as detailed in our methods under “Statistical Analysis”. In the time between the submission of our manuscript and the reviewer feedback, the most recent data from Senegal and Rwanda were released and accessible via the DHS’s API. For the 2019 DHS in Ethiopia, some modules are accessible from the API, but the Women’s Questionnaire (where the VAS question is probed) is not currently accessible and cannot be included.

We have redrawn all of our figures and tables with the most recent DHS data from Senegal and Rwanda.

References

1. Fawzi, W. W. & Wang, D. When should universal distribution of periodic high-dose vitamin A to children cease? *American Journal of Clinical Nutrition* **113**, 769–771 (2021).
2. Croft, T., Marshall, A. M., Allen, C. K., et al. *Guide to DHS Statistics*. (2018). Rockville, Maryland, USA: ICF.
3. United Nations Children’s Fund. *Coverage at a Crossroads: New directions for vitamin A supplementation programmes*. (2018). New York, New York, USA: UNICEF.